# Characterization of Double Leached Waelz Oxide for Identification of Fluoride Mineral

**Suchandra Sar** [1,*], **Lena Sundqvist Öqvist** [1], **Tobias Sparrman** [2], **Fredrik Engström** [1] **and Caisa Samuelsson** [1]

[1] Division Minerals and Metallurgical Engineering, Department of Civil, Environment and Natural Resources Engineering, Luleå University of Technology, 971 87 Luleå, Sweden; lena.sundqvist-oqvist@ltu.se (L.S.Ö.); fredrik.i.engstrom@ltu.se (F.E.); caisa.samuelsson@ltu.se (C.S.)

[2] Department of Chemistry, Faculty of Science and Technology, Umeå University, 901 87 Umeå, Sweden; tobias.sparrman@umu.se

\* Correspondence: suchandra.sar@ltu.se; Tel.: +46-920-491-013

**Abstract:** Double leached Waelz oxide (DLWO), with 76% zinc, is a secondary zinc containing raw materials obtained by the treatment of electric arc furnace dust. The content of fluoride in DLWO is still too high for direct leaching, as fluoride has a detrimental effect on electrowinning for zinc production. Knowledge of the characteristics of DLWO, and especially on how a fluoride mineral might exist, can contribute to further improvement of the selective leaching for the removal of fluoride. In this study, DLWO was characterized using analytical techniques, such as inductively coupled plasma-optical emission spectroscopy (ICP-OES), $^{19}F$ liquid-state nuclear magnetic resonance ($^{19}F$ LS NMR), X-ray powder diffraction analysis (XRD), scanning electron microscopy coupled with energy dispersive spectroscopy (SEM-EDS) and $^{19}F$ solid-state nuclear magnetic resonance ($^{19}F$ SS NMR). This study showed that DLWO mainly consisted of zincite ($ZnO$), cerussite ($PbCO_3$) and a spinel containing zinc, iron and manganese. The fluoride mineral identified was calcium fluoride ($CaF_2$). In SEM analysis, fluorine was found in larger grains together with calcium and oxygen, which was possibly calcium carbonate.

**Keywords:** metal recycling; zinc secondary dust material; characterization of double leached Waelz oxide; halogens; fluoride identification

---

## 1. Introduction

In 2017, 13.2 million metric tons of refined zinc metal were produced worldwide [1], of which 80% were obtained via the roasting, leaching and electrowinning (RLE) route. Zinc sulfide concentrates and some secondary materials roasted in fluidized bed furnaces form zinc oxide calcine with iron present as zinc ferrite. The calcine was leached for dissolution of zinc, and further purification was carried out before the electrolysis, e.g., selective precipitation and cementation [2].

Zinc, the third most important base metal next to aluminum and copper, is effective for protecting steel from corrosion. Globally, 50% of the produced zinc was used for galvanization of steel [1]. Galvanized steel available in different forms, such as sheets, pipes and wires, was used in the production of several household appliances, construction and automobile industries, etc. At the end of its life cycle, steel scrap was recycled in a scrap-based steel making, e.g., an electric arc furnace (EAF). Metallic zinc vaporized during the steelmaking process mainly report to the gas phase where zinc vapors are oxidized and condensed to form, e.g., zinc oxide if iron was present as zinc ferrite. Approximately, 15 kg of dust per ton of steel were generated during smelting in the EAF [3]. Zinc, iron, calcium and lead are the major elements present in EAF dust; hence, by recycling, the valuable

dust resources can be recovered, and at the same time, the need for landfill is reduced. Published data (Table 1) on dust from different steel plants indicated that its composition is site specific [4–15]. The type and composition of steel scrap and the process conditions determine the properties of the dust.

**Table 1.** Chemical composition of electric arc furnace (EAF) dust reported in the literature [4–15].

| Elements | Zn | Fe | Pb | Cd | Mn | Ca | CaO | F | Cl |
|---|---|---|---|---|---|---|---|---|---|
| Weight % | 4–33 | 18–49 | 1–5 | 0.01–0.15 | 0.4–6 | 1.0–10.0 | 0.4–0.7 | 0.01–0.88 | 0.01–7 |

Mineralogical analysis [4–6,15] on EAF dust showed that the predominant phases are spinels containing Fe, Zn, Mn, Cr, Ca, Mg and also zinc oxide. Moreover, chlorides of lead (Pb), fluorides of calcium and manganese (Ca and Mn), lead and calcium carbonates, as well as lead sulfate are reported. About 50–80% zinc existed as ZnO, and the rest was mainly associated with Fe as zinc iron ferrite spinel [12].

Pyro- and hydro-metallurgical methods for the recovery of zinc from EAF dust have been developed [7–10,13,14,16]. The treatment of EAF dust in the Waelz process at temperatures up to 1250 °C and in a $CO$–$CO_2$ atmosphere has been commercially established [16]. Reduced zinc evaporated and would be re-oxidized to form dust, which was collected in the upper part of the kiln. The dust known as Waelz oxide (WO) contained up to 60% zinc and was further purified for dehalogenation using a double-leaching process with sodium carbonate ($Na_2CO_3$) [17,18]. The double-leached WO (DLWO) contained 61–66% zinc, 7–9% lead, 3.5–5.5% iron and the content of the halogens was reduced to <0.1% for Cl and <0.15% for F in the leaching process [2]. The calcium distribution in DLWO was measured by transmission electron microscopy (TEM) [17], and it was indirectly concluded that fluoride was likely present as $CaF_2$. Using a similar method on water-washed EAF dust also indicated the presence of $CaF_2$ [15,19]. In this case, the crystalline structure was indirectly used for the identification of $CaF_2$ [19].

The halogen content has to be further reduced to enable direct leaching of zinc for electrowinning. Further, the removal of halogens can be done, e.g., by roasting together with zinc sulfide concentrates, or alternatively, fluoride in sulfuric leachate can be removed by selective precipitation [2]. The presence of chloride and fluoride as impurities in the electrolyte is a matter of concern in the electrowinning of zinc due to the possible $Cl_2$ gas formation at the anode and fluoride ions attack on the protected $Al_2O_3$ layer or metallic aluminum of the cathode. The fluoride attack may cause the formation of cavities in the cathode in which zinc can deposit. Additionally, the deposited zinc metal can form an alloy with aluminum metal, which causes difficulties to strip the zinc sheet.

In this work, the characterization of DLWO was carried out for the identification of minerals specifically fluoride species for a broader understanding of factors hindering the fluoride and chloride contents to be <0.01% enabling direct leaching of zinc. DLWO was characterized through chemical analysis, particle-size distribution, X-ray diffraction (XRD), scanning electron microscopy (SEM) coupled with energy dispersive spectroscopy (EDS) and $^{19}F$ solid-state magic-angle-spinning nuclear magnetic resonance ($^{19}F$ SS MAS NMR) spectroscopy.

## 2. Materials and Methods

### 2.1. Sample Preparation and Reagents

#### 2.1.1. Sample Preparation

The sample of DLWO used for the characterization study was supplied by Boliden AB, Odda, Norway. The received DLWO was divided into sub-samples using a riffle splitter, dried at 70 °C for 24 h before grinding with a mortar and pestle. Dried samples were stored in small sample bags under nitrogen-filled atmosphere inside a glass desiccator filled with silica gel.

2.1.2. Reagents

Reagents used in this work were aqua-regia freshly prepared by mixing nitric acid ($HNO_3$ 68%) and hydrochloric acid (HCl 37%) in 3:1 proportion, both purchased from VWR International S.A.S, Fontenay-sous-Bois, France. Silver nitrate ($AgNO_3$ EMSURE), sodium fluoride (NaF EMSURE) originated from Merck KGaA, Darmstadt, Germany; total ionic strength adjustment buffer (TISAB) and sodium peroxide ($Na_2O_2$ 97.7%) from VWR International bvba, Leuven, Belgium.

Reagents for LS (liquid state) [19]F NMR were Deuterium oxide ($D_2O$ 99.9 atom % D), Tetrafluoroacetic acid (TFA 99%) from Sigma-Aldrich Chemie Gmbh, Scnelldorf, Germany and internal standard sodium fluoride (NaF 99.995%) from AlfaAesar Puratronic Thermo Fisher- Kandel, GmbH, Karlsruhe, Germany. As External standards for [19]F SS MAS NMR studies cadmium fluoride ($CdF_2$, 99.99%), potassium fluoride (KF, 99.99%), sodium fluoride (NaF, 99.995%), lead (II) fluoride ($PbF_2$, 99.997%), zinc fluoride ($ZnF_2$, 99.995%), magnesium fluoride ($MgF_2$, 99.99%) and calcium fluoride ($CaF_2$, 99.985%), highest available purity grade chemicals were purchased from AlfaAesar Puratronic(Thermo Fisher- Kandel, GmbH, Karlsruhe, Germany).

*2.2. Particle Size*

Particle size distribution of DLWO was conducted through laser-diffraction measurements using a CILAS 1064 unit (CILAS, Orleans, France) and Fraunhofer approximation was used for the evaluation of the collected data.

*2.3. Elemental Analysis*

The elemental analysis was conducted with inductively coupled plasma-optical emission spectroscopy (ICP-OES, Model: Thermo Scientific ICAP 7200 DUO, Thermo Fisher Scientific, Waltham, MA, USA) following a standard method. For the analysis of the metal contents, the sample was digested in aqua-regia at 100 °C for 30 min and the diluted sample was analyzed according to European standard EN 13346:2000.

For chloride determination, the sample was heated to 50 °C in a 4 M $HNO_3$. The filtered leachate was diluted up to 100 mL, and titrimetric determination of chloride was conducted using a 0.05 M $AgNO_3$ solution as titrant (standard method 1346601 AB-130 3 En). Metrohm Titrando 888 titrator unit with the tiamo software and Metrohm Ag titrode electrode 6.0430.100 were used.

For a quantitative fluoride analysis with the fluoride-ion selective electrode (FISE Metrohm) and LS (liquid state) [19]F NMR, a known amount of sample was fused with $Na_2O_2$ [20]. The fused sample was dissolved in hot water to a volume of 100 mL. The obtained leached liquor was analyzed for fluoride using FISE (Metrohm FISE 6.0502.150, Herisau, Switzerland, Titrando 888 with tiamo software), and the standard methods for the determination of fluoride were applied [21]. TISAB was added in order to avoid interference from iron. In the case of LS [19]F NMR, the obtained leached liquor was freeze-dried and further dissolved in 10 mL of $D_2O$.

The LS [19]F NMR experiments were recorded on a 400 MHz Bruker Avance III spectrometer equipped with a broadband [19]F (BBF) probe (BRUKER BioSpin GMBH, Rheinstetten, Germany). The F quantification was done using a known amount of TFA as an internal standard as well as NaF solution in $D_2O$ of known concentration as an external standard. NMR has an inherent analytical characteristic in which the signal intensity in the NMR spectrum is directly proportional to the number of nuclei responsible for a particular resonance [22]. By comparing the relative peak area [23] under the standard TFA peak and the sample fluoride peak in the [19]F NMR spectrum, the amount of fluoride in the sample was determined using the method of internal standard. In the method of external standard, the peak areas under each recorded [19]F NMR spectra for a series of NaF solution of different known concentrations were determined. To obtain the standard curve, the peak areas were plotted against the corresponding concentrations. The area under the fluoride peak in the sample of DLWO was determined, and the fluoride concentration was estimated from the standard curve. Spectra were

collected with 16 transients, and TopSpin software (version 3.5, BRUKER BioSpin GMBH, Rheinstetten, Germany) was used to process and analyze the NMR data sets.

All elemental analysis tests were done in triplicate and the results were presented as a mean value.

### 2.4. Wet High-Intensity Magnetic Separators (WHIMS)

To enable the determination of fluoride compounds by [19]F SS NMR, the contents of magnetic and paramagnetic species had to be reduced; thus the magnetic separation was done to obtain a fraction with low concentration in these elements. A water slurry of DLWO having 20% pulp density was directly fed into a wet high-intensity magnetic separator (Jones separator) by applying a magnetic field of 6 amps (4650 Gauss) and current of 210 Volts. The magnetic and non-magnetic fractions were collected separately and dried at 100 °C for 48 h before further analysis. The nonmagnetic (DLWO-NM) and magnetic (DLWO-Mag) fractions, as well as the original DLWO, were analyzed chemically, measured with XRD, subjected to NMR studies and SEM analysis.

### 2.5. Mineralogical Characterization

XRD was conducted using a PANalytical Empyrean X-ray diffractometer (Malvern Panalytical, Almelo, The Netherlands), equipped with copper Kα radiation of 45 kV and 40 mA. The XRD pattern was recorded in the 2-theta range of 10–90° with a step of 0.026°. Evaluation of data was carried out using the software HighScore Plus (version - 4.7, PANalytical B.V., Almelo, The Netherlands).

SEM was conducted on the polished and carbon-coated epoxy sample using Zeiss Gemini Merlin equipped with an energy dispersive spectrometer (EDS- Xmax 80 mm, Zeiss, Oberkochen, Germany). The acceleration voltage was set to 20 keV, and the emission current was 1.0 nA.

[19]F SS MAS NMR was conducted using a 500 MHz Bruker Avance III spectrometer (BRUKER BioSpin GMBH, Rheinstetten, Germany) equipped with a 4 mm MAS probe. MAS generates rotational echoes in the time domain (free induction decay, FID) that Fourier-transforms into the spinning sideband separated by the rotation frequency. The central band reveals the isotropic shift (and always appears at the same position in the spectra irrespective of the rotor speed), while the spinning sideband patterns contain information on the anisotropy in the interaction. Their intensity and positions are dependent on the speed of the rotor [24]. The [19]F spectra were acquired at 13 kHz spinning using a direct polarization pulse of 1 μs corresponding to about 15° excitation angle followed by a rotor-synchronized delay of 75 μs in order to start the acquisition of the FID at the first rotational echo and thus minimize baseline distortions. A recycle delay of 30 s was used, and the [1]H decoupling was not applied. In order to identify the central band from the spinning sidebands, the [19]F spectra were also recorded with a 10 kHz spinning speed.

## 3. Results and Discussion

### 3.1. Particle Size

The particle size distribution of the examined DLWO is given in Figure 1. It was found that 50% of particles were <2.20 μm whereas the majority (90%) of the particles were <20 μm. The highest size population density was at ~1.7 μm, while there was another maximum size population density centered at ~17 μm. All particles were less than 60 μm in size.

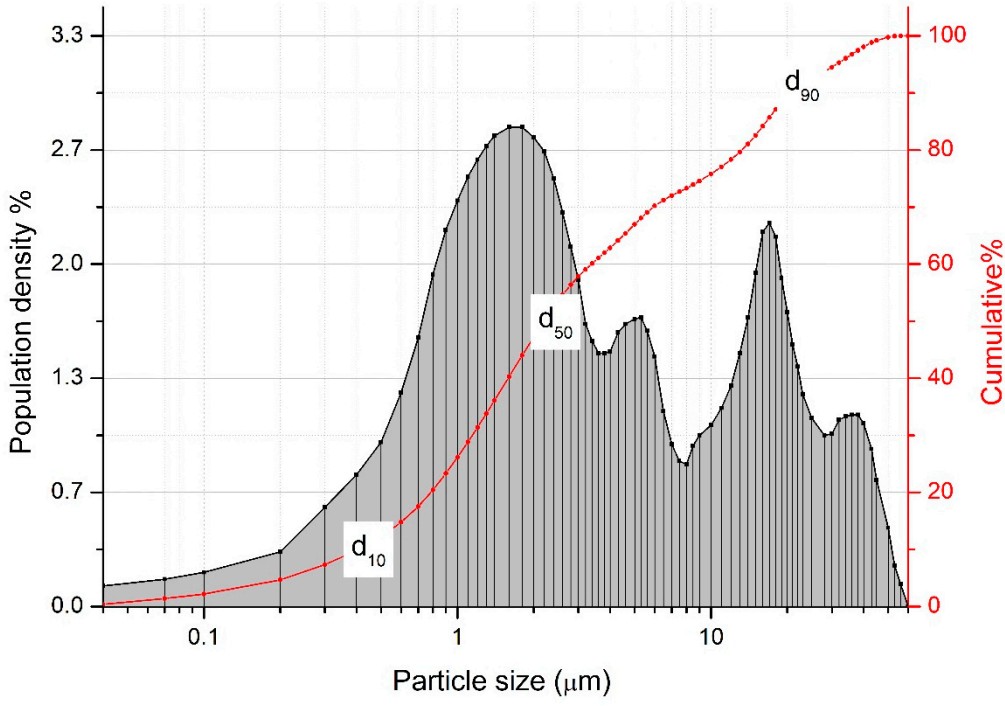

**Figure 1.** Particle size distribution of DLWO.

### 3.2. Elemental Analysis and Magnetic Separation

Elemental analysis showed that DLWO mainly consisted of zinc; the sample material used in this study had comparably high zinc content. Lead and iron were present in significant amounts, and other elements are present in a minor amount, cf. Table 2. The fluoride contents estimated using FISE and $^{19}$F LS NMR were in agreement and found to vary between 0.10–0.15% in individual samples. There was no indication of iron interference during the fluoride measurement as the NMR spectra did not show any interference by iron. The chloride concentration determined by potentiometric titration was found to be 0.16%. Elemental concentration in DLWO, the non-magnetic fraction (DLWO-NM) and magnetic fraction (DLWO-Mag) are shown Table 2.

**Table 2.** Chemical composition in wt% of DLWO, DLWO-Non magnetic (NM) and DLWO- Magnetic (Mag).

| Elements | Zn | Pb | Fe | Ca | Cd | Mn | Mg | F | Cl |
|----------|------|------|------|------|------|------|------|------|---------|
| **DLWO** | 76.1 | 1.58 | 1.46 | 0.80 | 0.17 | 0.18 | 0.13 | 0.10 | 0.16 |
| **DLWO-NM** | 76.6 | 1.59 | 1.17 | 0.68 | 0.17 | 0.15 | 0.11 | 0.09 | <0.010 |
| **DLWO-Mag** | 63.4 | 1.16 | 8.03 | 3.36 | 0.21 | 0.79 | 0.44 | 0.38 | <0.010 |

The magnetic separation experiment was repeated four times, and the average values with standard deviations are shown in Figure 2. Based on mass balance, approximately 3% of zinc and lead, 24% of iron and 15–20% of F, Ca and Mn reported to the magnetic fraction cf. Figure 2.

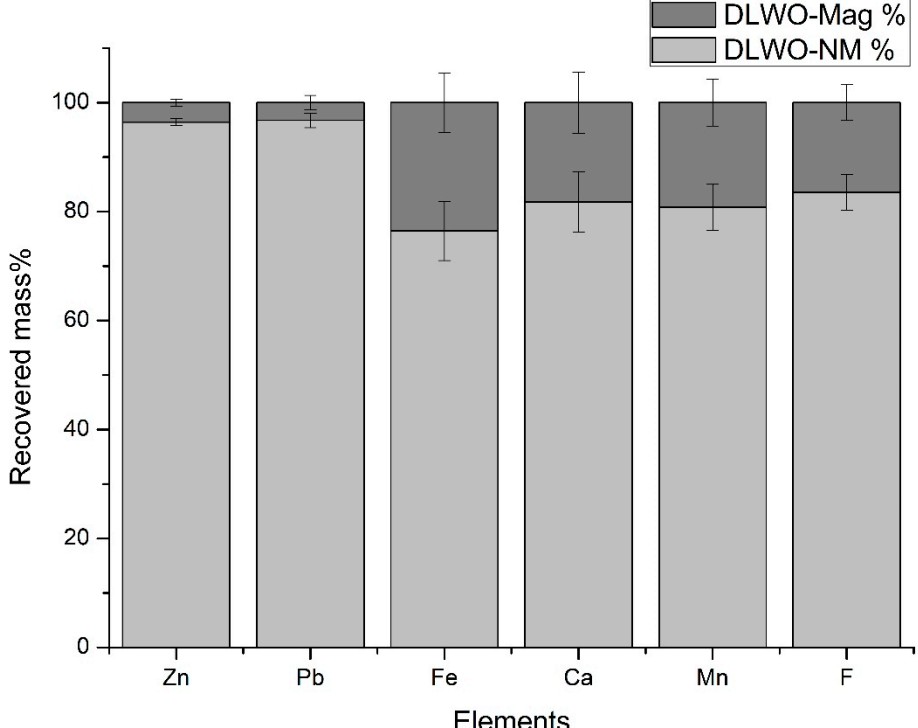

**Figure 2.** Distribution with standard deviation deduced from the mass-balance calculation for selected elements in DLWO-NM and DLWO-Mag.

*3.3. XRD*

The XRD data (Figure 3) showed that the major phases in the sample of DLWO and DLWO-NM were zincite (ZnO) and cerussite ($PbCO_3$), which were in agreement with the reported results from other studies [25]. $PbCO_3$ was formed during the soda washing of WO. XRD analysis on DLWO-Mag also showed a spinel containing zinc, iron and manganese, along with zincite and hydrocerussite (lead carbonate hydroxide—$Pb_3(CO_3)_2(OH)_2$). The results of the chemical analysis (Table 2) and XRD diffractogram showed that iron and manganese were enriched in the magnetic fraction. Hydrocerussite might have been formed during the treatment of DLWO by magnetic separation as the sample was subjected to a large amount of water when making a slurry. Afterward, the obtained fractions were dried at 100 °C. Halide-containing species were not identified by XRD due to the low contents of fluoride and chloride.

*3.4. SEM*

SEM-EDS study showed that DLWO was a heterogeneous material mainly containing zinc and lead with small amount of calcium and iron. Observation of chloride and fluoride was limited to a few specific grains. In the bulk of the sample, zinc, lead, oxygen and traces of calcium coexisted in some spots. SEM indicated that one phase was most likely zinc oxide with traces of other elements (marked 1 in Figure 4), and another phase was dominated by lead and oxygen (marked 2 in Figure 4) that could correspond to lead carbonate. High content of calcium was detected in some larger-size grains, where fluorine was also detected in some spots (marked by a red outline in Figure 4). The analysis indicated the presence of oxygen in these grains. Based on this analysis and the knowledge on the sodium-carbonate washing of Waelz oxide, these grains possibly consisted of a mix of calcium fluoride and calcium carbonate. However, the presence of calcium hydroxide cannot be excluded.

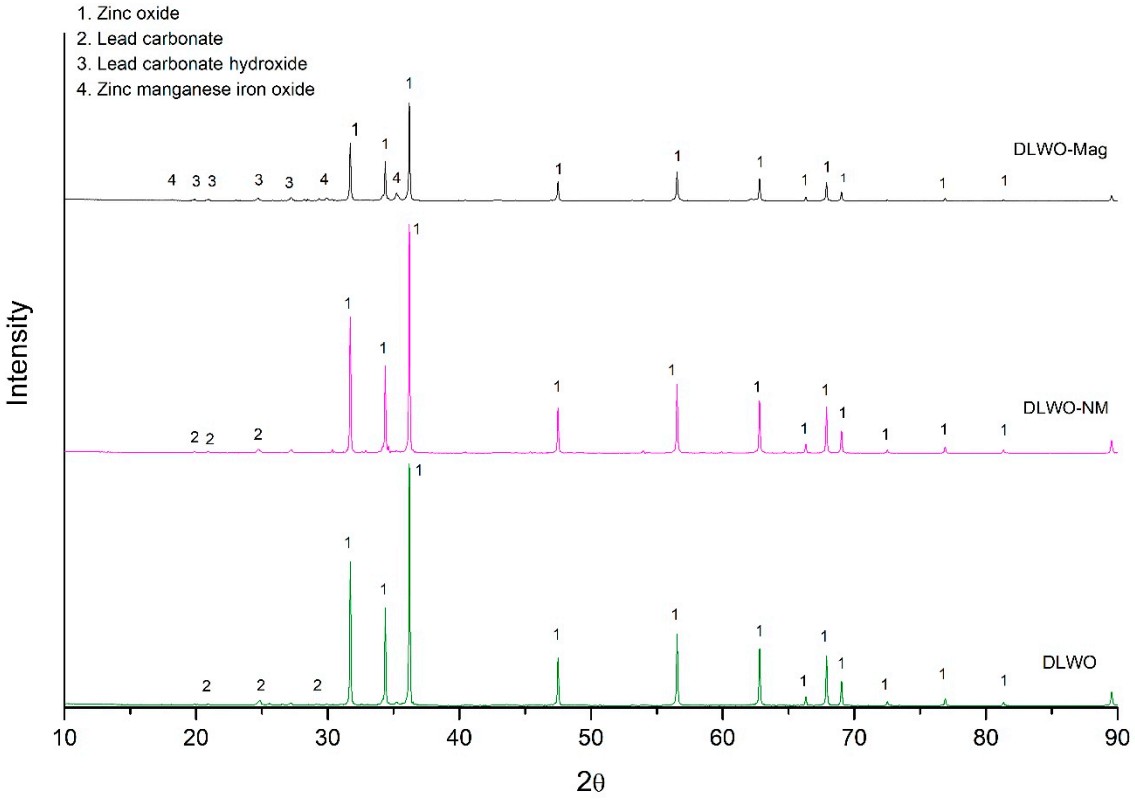

**Figure 3.** The XRD diffractogram of DLWO, DLWO-NM and DLWO-Mag.

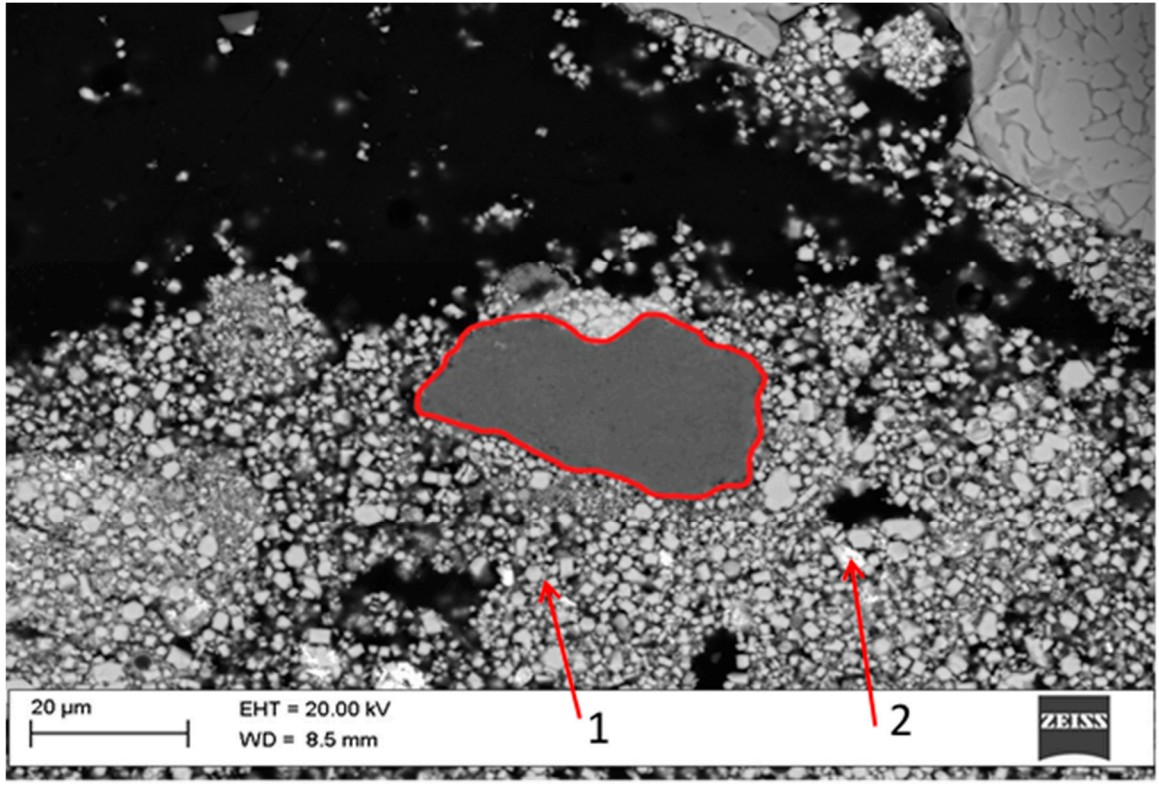

**Figure 4.** SEM image of DLWO. Atomic % range of F is 6.81–36.39 and of Ca, 26.85–40.24.

### 3.5. $^{19}F$ SS-MAS FT-NMR

At a high rotational speed of 13 kHz, the $^{19}F$ SS-MAS FT-NMR spectra of standard fluoride compounds generated one single peak, except for $PbF_2$ and $ZnF_2$ that gave double central peaks, which are in agreement with the literature values [26], cf. Table 3.

**Table 3.** Recorded central-band $^{19}F$ shifts of standard fluoride compounds.

| Shift (ppm) | NaF | CaF$_2$ | MgF$_2$ | ZnF$_2$ | PbF$_2$ | KF | CdF$_2$ |
|---|---|---|---|---|---|---|---|
| | −224 | −108 | −198 | −182 | −19 | −133 | −194 |
| | - | - | - | −202 | −58 | - | - |

The NMR spectra of DLWO, DLWO-NM and standard salt calcium fluoride is presented in Figure 5. Untreated DLWO NMR spectra showed the generation of several spinning sidebands due to the inherent $^{19}F$ chemical shift anisotropy as well as the paramagnetic shift anisotropy and relaxation enhancement caused by the paramagnetic iron [27] and manganese species present in the sample. The clarity of the spectra (Figure 5) was significantly improved by reducing these paramagnetic species in DLWO-NM.

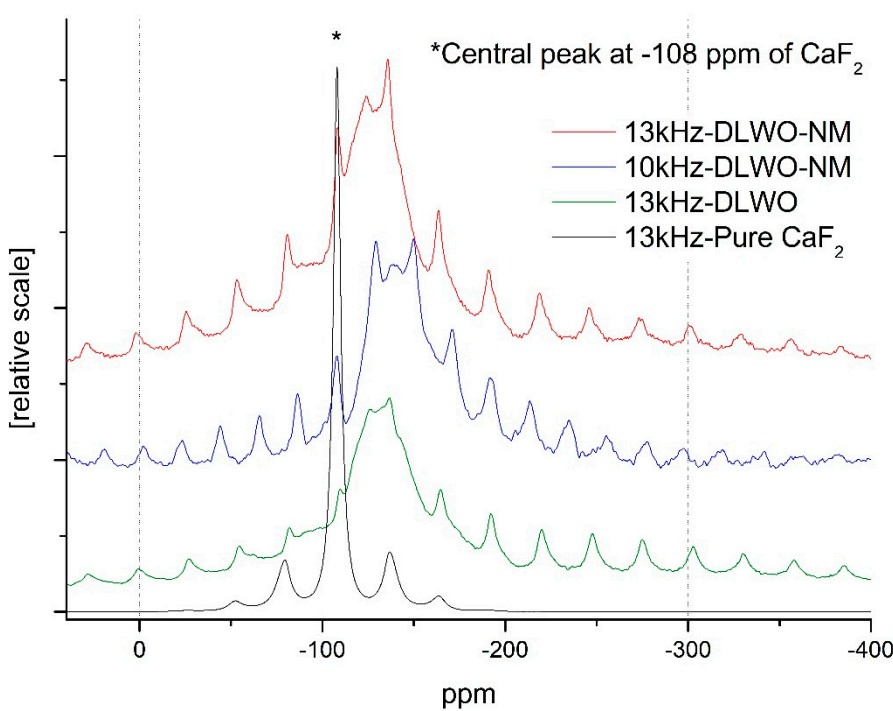

**Figure 5.** $^{19}F$ solid-state magic angle spinning nuclear magnetic resonance ($^{19}F$ SS MAS NMR) spectra (qualitative analysis).

The intense broad hump centered at about −120 ppm was the probe background. The overlay of the DLWO-NM run at 10 kHz and 13 kHz spinning speeds (Figure 5) showed that all peaks (sidebands) shifted, except the one at −108 ppm, which is the central band. It was found that the sample peak and the standard $CaF_2$ peak matched while none of the other standard peaks was close to the sample peak. This confirmed the presence of $CaF_2$ in the non-magnetic fraction of DLWO. The remaining fluoride of 0.08% in DLWO-NM was enough for identification using $^{19}F$ SS NMR as $^{19}F$ has a high magnetogyric ratio and 100% natural abundance together yielding a high NMR sensitivity. None of the $^{19}F$ spectra presented in this article were background-corrected by subtracting empty rotor background from the acquired spectra. Spectra for all reference compounds had such strong signals, so that the probe background was almost invisible and thus not needed to be subtracted. For the

DLWO spectra, the $CaF_2$ peaks were near but not exactly at the same position as the background peak; thereby, mathematical operators (e.g., subtraction) were avoided as they might generate some artifacts.

Chloride mineral phases were not identified in this characterization work by any of the above-mentioned analytical tools. Chemical analyses of chloride in DLWO, DLWO-Mag and DLWO-NM indicated that some chlorines were removed by the water during WHIMS separation.

Approximately, 0.10–15% fluoride remained in DLWO after the double-leaching of Waelz oxide.

The characterization indicated that fluoride was mainly present as calcium fluoride together with calcium carbonate in larger grains. It was possible that $CaF_2$ is co-precipitated with $CaCO_3$ during the sodium-carbonate washing of Waelz oxide [28].

Studies on pure calcium fluoride could give further understanding on how the mechanisms hindering the full removal of fluoride from Waelz oxide are linked to the co-precipitation of calcium fluoride with calcium carbonate or the dissolution of the calcium fluoride itself.

## 4. Conclusions

- The characterization of DLWO showed that 90% of the particles were <20 μm and mainly consisted of zincite ($ZnO$), cerussite ($PbCO_3$) and spinel with zinc, iron and manganese.
- The only identified fluoride mineral was $CaF_2$ in the non-magnetic fraction of DLWO.
- Fluoride was mainly present together with calcium and oxygen in larger grains.

**Author Contributions:** Conceptualization, C.S.; methodology, S.S., F.E. and T.S.; formal analysis, S.S.; investigation, S.S.; data curation, S.S.; writing—original draft preparation, S.S.; writing—review and editing, S.S., C.S., L.S.Ö. and T.S.; supervision, C.S., F.E. and L.S.Ö.; project administration, C.S., L.S.Ö.; funding acquisition, C.S.

**Funding:** This research was funded by Boliden Commercial AB through Bolidenpaketet.

**Acknowledgments:** Helpful discussions regarding magnetic separation with Associate Professor Bertil Pålsson is greatly appreciated. Active support in the project by NMR for Life (Swedish NMR Centre) is highly appreciated by the authors. Fruitful discussions with Boliden is highly acknowledged.

**Conflicts of Interest:** The authors declare no conflicts of interest.

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
