# Peer review of "Characterization of Double Leached Waelz Oxide for Identification of Fluoride Mineral"

_metals, doi:10.3390/met9030361_

Round 1

Reviewer 1 Report

The research presented in this work is very interesting from the point of view of the technology of zinc production using hydrometallurgical method. But the work should include additional information such as:

1. from where in the steelmaking dust such a large amount of F and Cl

2. what are the acceptable concentrations of F and Cl in the WO that is intended for the leaching process

explanations 1&2 should be included in the introduction

3. what was the purpose of magnetic separation, does it have any practical (technological) aspect?

4. Table 3 indicates that DLWO-mag contains 0.38 wt. % of F. From this point of view, the material may be directed to leaching again to reduce the concentration of fluoride - is it possible?

explanations 3&4 should be included in the discussion

lines 190-199 - where are the results of analysis ??? - there is only SEM image

Author Response

Response to Reviewer 1 Comments

Point 1: from where in the steelmaking dust such a large amount of F and Cl

Response 1:  Secondary source- steel scrap

CaF2 is also used as flux

Point 2: what are the acceptable concentrations of F and Cl in the WO that is intended for the leaching process

Response 2:  Mentioned in lines 75-80

Point 3: what was the purpose of magnetic separation, does it have any practical (technological) aspect?

Response 3:  To conduct NMR analyses magnetic particles has to be minimized as these disturb the NMR signal. Magnetic separation was performed for preparation of suitable sample for NMR.

Point 4: Table 3 indicates that DLWO-mag contains 0.38 wt. % of F. From this point of view, the material may be directed to leaching again to reduce the concentration of fluoride - is it possible?

Response 4:  Magnetic separation was done as only part of characterization study for preparing suitable sample for NMR studies.

Point 5: lines 190-199 - where are the results of analysis ??? - there is only SEM image

Response 5:  Information added below the figure 4. Kindly see the attached word document. Several SEM analysis were performed.

Reviewer 2 Report

The manuscript can be published in its current form. My reasons are as follows:

i.I consider that the manuscript is of high quality
ii. In my opinion, the topic of work is novel
iii. The results serve to improve the field of recycling of EAF dusts.

The leaching of the WO can also be carried out with ammonium carbonate. In this case F2Ca would not be produced, What do the authors think?

Author Response

Response to Reviewer 2 Comments

Point 1: The leaching of the WO can also be carried out with ammonium carbonate. In this case F2Ca would not be produced, What do the authors think?

Response 1:  (NH4)2CO3 should act as similar leaching reagent like Na2CO3. Both enhances the dissolution and minimize the presence of CaF2.